# Oral Cancer-Derived miR-762 Suppresses T-Cell Infiltration and Activation by Horizontal Inhibition of CXCR3 Expression

**DOI:** 10.3390/ijms26031077

**Published:** 2025-01-26

**Authors:** Hsuan-Yu Peng, Chia-Wei Chang, Ping-Hsiu Wu, Li-Jie Li, Yu-Lung Lin, Michael Hsiao, Jang-Yang Chang, Peter Mu-Hsin Chang, Hsin-Lun Lee, Wei-Min Chang

**Affiliations:** 1School of Oral Hygiene, College of Oral Medicine, Taipei Medical University, Taipei 110, Taiwan; a03776@tmu.edu.tw; 2Research Center of Oral Translational Medicine, College of Oral Medicine, Taipei Medical University, Taipei 110, Taiwan; 3TMU Research Center of Cancer Translational Medicine, Taipei Medical University, Taipei 110, Taiwan; jychang@tmu.edu.tw; 4Division of Family Dentistry, Department of Dentistry, Taipei Medical University Hospital, Taipei 110, Taiwan; 153054@h.tmu.edu.tw; 5Department of Radiology, School of Medicine, College of Medicine, Taipei Medical University, Taipei 110, Taiwan; wupinghsiu@tmu.edu.tw; 6Graduate Institute of Clinical Medicine, College of Medicine, Taipei Medical University, Taipei 110, Taiwan; 7Department of Radiation Oncology, Taipei Medical University Hospital, Taipei 110, Taiwan; 8TMU Proton Center, Taipei Medical University, Taipei 110, Taiwan; 9Ph.D. Program of School of Dentistry, College of Oral Medicine, Taipei Medical University, Taipei 110, Taiwan; xu3xu4ru56lily@gmail.com; 10Department of Oral Pathology, Graduate School of Dentistry, Osaka University, Osaka 565-0871, Japan; 11The Ph.D. Program for Translational Medicine, College of Medical Sciences and Technology, Taipei Medical University, Taipei 110, Taiwan; yllin4671@tmu.edu.tw; 12Genomics Research Center, Academia Sinica, Taipei 115, Taiwan; mhsiao@gate.sinica.edu.tw; 13Department of Oncology, Taipei Veterans General Hospital, Taipei 112, Taiwan; ptchang@nycu.edu.tw; 14Faculty of Medicine, College of Medicine, National Yang Ming Chiao Tung University, Taipei 112, Taiwan; 15Institute of Biopharmaceutical Sciences, National Yang Ming Chiao Tung University, Taipei 112, Taiwan

**Keywords:** OSCC, microRNA horizontal transmission, CXCR3, tumor immune escaping

## Abstract

Oral squamous cell carcinoma (OSCC) is an immune-cold tumor characterized by an immunosuppressive microenvironment with low cytotoxic activity to eliminate tumor cells. Tumor escape is one of the initial steps in cancer development. Understanding the underlying mechanisms of cancer escape can help researchers develop new treatment strategies. In this study, we prove the oral oncogenic miR-762 can suppress T-cell recruitment and cytotoxic activation in the tumor microenvironment (TME) through horizontal transmission from OSCC cells to adaptive immune T cells. Public database analysis and quantitative real-time polymerase chain reaction (qRT-PCR) were used to determine the prognosis and expression of miR-762 in OSCC. T-cell activation by flow cytometry, qRT-PCR, IL-12 secretion, and T-cell recruitment and cytotoxicity abilities were conducted in the miR-762 manipulation T-cell and OSCC-T-cell co-culture system. A luciferase reporter and CXCR3 protein expression were also carried out to validate the direct interaction between CXCR3 and microRNA (miR)-762. This horizontal transmission of miR-762 directly suppresses CXCR3 expression in T cells, inhibiting CXCR3-induced T-cell migration and downstream T-cell cytotoxic activity by disrupting AKT activation. Additionally, miR-762 transmission suppressed T-cell activation marker expression, T-cell proliferation, IL-12 secretion, and T-cell cytotoxicity. In conclusion, our findings reveal a novel miR-762/CXCR3 axis that regulates the immunosuppressive microenvironment in OSCC and may be a potential RNA-targeted therapeutic approach to restore the anti-tumor immune response in OSCC treatment.

## 1. Introduction

Cancer is the deadliest disease, with an estimated 18.1 million new cases and 10 million deaths annually. In the USA, 1.9 million new cases and 609,820 deaths are expected each year [1]. Oral cancer is severely impacting patients’ quality of life and causing socioeconomic impacts [2]. The host immune system plays a crucial role in eliminating cancer cells during tumor initiation [3]. Effector immune cells, like T cells, γδT cells, and natural killer (NK) cells, are responsible for elimination [4]. These cells must infiltrate into the tumor microenvironment (TME). However, OSCC employs several strategies to evade the immune system, including expression of inflammatory cytokines, suppression of cytotoxic CD8 lymphocytes, downregulation of antigen processing machinery, generation of specific inhibitory lymphocytes, and expression of immune checkpoint ligands and receptors. These mechanisms increase OSCC resistance to cytotoxic T cells and inhibit T-cell function, facilitating tumor initiation and growth [5]. Understanding immune-escape mechanisms is essential for developing new treatment strategies for OSCC.

Chemokines are small cytokines that direct immune cell recruitment. The chemokine family consists of about 50 ligands, 20 G protein-coupled receptors (GPCRs), and 4 atypical chemokine receptors (ACKRs) [6]. They play a key role in cancer immunity by attracting immune cells to the tumor microenvironment (TME). CXCR3, a critical receptor, aids in T and NK cell recruitment and macrophage polarization [7]. Its expression is linked to the progression of cancers like breast, colorectal, and pancreatic. High levels of CXCR3 ligands in pancreatic and colorectal cancers are associated with poor survival, making it a potential prognostic marker [8]. The diverse functions of CXCR3 underscore its potential as a target for immunotherapy and its significance in cancer and immune regulation.

The TME is a dynamic ecosystem involving cancer cells and host cells like immune cells, fibroblasts, and endothelial cells [9]. In tumor initiation, dysregulated immune cell distribution shapes the TME immune landscape [10]. TME immune status affects patient prognosis and the outcomes of treatment. The TME EVs harbor proteins and free nucleic acids, such as enzymes, receptors, mRNA, non-coding RNA, and microRNA, and serve as colonial rulers for horizontal transmission between cancer donors and host recipient T cells [11]. MicroRNAs, known for their stability and gene regulation capabilities, play a critical role in TME [12]. Aberrant microRNA expression has been reported in OSCC tumorigenesis and progression [13]. miR-762, an oncogenic microRNA, contributes to OSCC and is a poor prognostic marker in colon cancer due to WNT signaling [14]. Previously, we found that miR-762 is a serum prognostic marker and contributes to colon cancer progression through WNT signaling [15]. Here, we found that miR-762 suppresses the immune response in OSCC by targeting CXCR3 and hindering cytotoxic T-cell recruitment and activation.

## 2. Results

### 2.1. miR-762 Significantly Associated with the Poor Prognosis of OSCC

To investigate the relationship between miR-762 and clinical manifestations in OSCC patients by using the Kaplan–Meier plotter database. The high-miR-762 group had significantly shorter overall survival (OS, Figure 1A). The median OS was 77.3 months in the low-miR-762 group versus 37.77 months in the high-miR-762 group. We also found that miR-762 levels are high in OSCC cell lines compared to normal keratinocyte cells (Figure 1B). OSCC is an immune cold tumor, and OSCC typically has an immunosuppressive microenvironment, involving immune checkpoint proteins, environmental immunosuppressive factors, and an imbalance between immunosuppressive and effective immune cell compositions [3]. This leads to diminished T-cell numbers and activity in the tumor, resulting in poor prognosis for OSCC patients [16]. We found that miR-762 is secreted by OSCC cells (Figure 1C), with Cal-27 having the lowest secretion of miR-762 among the five OSCC cell lines (Cal-27, SCC-4, HSC-3, SAS, and SCC-15). Secreted miRNAs are highly stable in the TME and may serve as critical intercellular communication molecules, allowing cancer cells to manipulate or diminish surrounding recipient T cells through horizontal transfection [17]. miR-762 serves as an exosomal microRNA from OSCC cells (Appendix A). Furthermore, we found that miR-762 not only suppressed NF-κB signaling transactivation (Figure 1D) but also E-cadherin promoter transcription activity (Figure 1E), which may inhibit immune cell maturation and prevent migration into the TME. These results suggest that miR-762 plays an important role in the OSCC-TME with immune suppression function. We speculate that miR-762 may lead to decreased T-cell activation and function in the OSCC-TME.

### 2.2. Exogenous miR-762 Inhibits the Jurkat Cells Activation and Migration Ability

To demonstrate the function of extracellular miR-762 in the OSCC-TME, particularly in tumor immune evasion through the prevention of T-cell activation, we transfected Jurkat cells with miR-762-PM for 72 h. The surface T-cell activation markers, CD25 and CD69, were inhibited by miR-762 transfection (Figure 2A). Additionally, the proinflammatory cytokine IL-12, which augments the growth, differentiation, and activation of cytotoxic CD8 T cells [18], was reduced by miR-762 manipulation compared to the NC control (Figure 2B). This indicates that miR-762 inhibited Jurkat-cell cytotoxic activity.

Furthermore, miR-762 transfection reduced Jurkat cell proliferation (Figure 2C) and Transwell migration ability (Figure 2D). These results show that miR-762 is an important suppressor of T-cell migration and activation in the TME. Additionally, we introduced miR-762 into Cal-27 cells, which have both lower intracellular and secretion levels of miR-762, and harvested the cell culture supernatant as an OSCC-derived conditioned medium to mimic the TME. The miR-762-containing Cal-27 conditioned medium (Cal-27-miR-762-CM) also suppressed PHA-activated Jurkat cell migration and further inhibited the gene expression of T-cell activation markers such as IL2RA, CD69, and CD71 (Figure 2F). Taken together, these results indicate that miR-762 in the TME can inhibit T-cell recruitment and activity.

### 2.3. Exogenous miR-762 Inhibits T-Cell CXCR3 Expression

CXCR3 is primarily expressed in T cells, dendritic cells, and natural killer cells, playing a crucial role in T-cell trafficking and function [19]. Moreover, CXCR3 has three variants, A and B, which exhibit differences in their expression profiles in the TME and have different functions in cancer progression [20]. TargetScan analysis predicted a direct binding interaction between miR-762 and CXCR3 (Figure 3A). We performed a CXCR3-3′-UTR reporter assay with or without the miR-762 binding site mutation. The CXCR3-3′-UTR was significantly suppressed by miR-762 mimic (PM) transfection but was abolished by the miR-762 binding site mutation (Figure 3B). Additionally, ectopic miR-762 expression directly suppressed CXCR3 mRNA and protein expression in Jurkat cells (Figure 3C–E). These results indicate that miR-762 could regulate CXCR3 through both reducing mRNA stability and protein translational inhibition. Moreover, miR-762 selectively reduced the expression of the T-cell proliferation and migration-promoting variant CXCR3A.

To further examine the role of extracellular miR-762 in the OSCC tumor microenvironment, we applied an OSCC-conditioned medium to Jurkat cells to assess CXCR3 expression. The conditioned medium harvested from miR-762-expressing Cal-27 cells also suppressed CXCR3 expression in Jurkat cells (Figure 3G,H). These results indicate that OSCC-secreted miR-762 can suppress T-cell CXCR3 expression in the oral tumor microenvironment. Both direct transfection of miR-762 and OSCC-derived miR-762 conditioned medium were found to stimulate the expression of the CXCR3B variant, which is generated through alternative splicing (Figure 3F–H). Unlike CXCR3B, which exhibits tumor-suppressive functions, CXCR3A promotes tumorigenesis by enhancing cell proliferation and migration. The regulation of CXCR3B involves unique transcription factors, NRF2 and BACH1 [21], which modulate its expression and downstream effects in response to miR-762. This suggests that miR-762 may influence the alternative splicing or transcriptional regulation of CXCR3 isoforms, thereby shaping their distinct roles in tumor progression and immune modulation. Additionally, CXCR3B functions as a receptor for CXCL4, a chemokine known to inhibit cytotoxic T-cell migration and activation [22]. These findings suggest that miR-762 may exert a unique T-cell suppression function through the CXCL4/CXCR3B axis, distinct from the tumor-promoting role of CXCR3A.

### 2.4. CXCR3 Promotes Cytotoxic T-Cell Activation

To further examine whether CXCR3 contributes to T-cell activation, we evaluated the T-cell activation status following ectopic CXCR3 expression (Figure 4A). We found that CXCR3 expression enhanced the expression of T-cell activation genes such as IL2RA (CD25), CD69, and CD71 mRNA (Figure 4B). Additionally, we observed that CXCR3 expression increased the expression of T-cell activation surface markers CD69, CD25, and CD154 in Jurkat cells (Figure 4C). Moreover, we also found that CXCR3 increased Jurkat T-cell proliferation (Figure 4D). IL-12, a T cell-stimulating cytokine, promotes the growth and function of T cells and the production of interferon-gamma (IFN-γ) and tumor necrosis factor-alpha (TNF-α) from T cells and natural killer (NK) cells [23]. CXCR3 expression also stimulated IL-12 secretion from Jurkat cells (Figure 4E).

Taken together, our results show that CXCR3 enhances Jurkat T-cell cytotoxicity by promoting T-cell polarization and proliferation and increasing IL-12 secretion, thereby proving that CXCR3 plays a crucial role in T-cell activation.

### 2.5. miR-762/CXCR3 Suppression Axis Is Critical in Environment T-Cell Activation

The molecular mechanisms underlying CXCR3 regulation of T-cell function are not fully understood. Several studies have suggested that CXCR3 signaling involves Ras/ERK, Src, and PI3K/AKT pathways [24]. Additionally, CXCL10/CXCR3 activation can stimulate PI3K/AKT signaling, thereby promoting Th1 cell differentiation and migration. The PI3K/AKT pathways are then stimulated, leading to the promotion of cell survival, proliferation, and cytotoxic activity of memory T and NK cells.

To clarify the interplay among the miR-762/CXCR3/Akt axis in OSCC immune evasion, we investigated whether OSCC miR-762-induced CXCR3 downregulation plays a critical role in T-cell activation. To this end, we generated a CXCR3 expression plasmid lacking the 3′-UTR sequence, thereby preventing inhibition by miR-762. In the miR-762 mimic-transfected Jurkat cells, restoring CXCR3 expression (Figure 5A) could restore AKT activity (Figure 5B). The restored CXCR3 expression further reactivated T-cell polarization (Figure 5C,D) and proliferation ability (Figure 5E). Moreover, the critical T-cell cytokine IL-12 secretion was also restored by CXCR3 expression (Figure 5F).

In summary, these results suggest that the miR-762/CXCR3 axis may regulate T-cell activation, polarization, proliferation, and IL-12 secretion in the tumor microenvironment.

### 2.6. Horizontal Transmission of miR-762 Promotes Immune Escape in OSCC

Finally, we aimed to demonstrate the horizontal transmission of miR-762 between OSCC cells and T cells, which regulates T-cell cytotoxicity and promotes OSCC immune escape. To do this, we used an indirect coculture system (Figure 6A). When miR-762 was introduced into low-expressing Cal-27 cells, T-cell migration (Figure 6B) and cytotoxicity (Figure 6C) were abolished by miR-762 expression. Moreover, this cancer immune escape could be reversed by reintroducing CXCR3 into the T cells.

Taken together, these results suggest that miR-762 attenuates T-cell cytotoxicity against cancer by reducing the expression levels of CXCR3. These findings indicate that T-cell inactivation induced by OSCC is primarily mediated by OSCC-derived miR-762.

## 3. Discussion

OSCC employs various immune-escape mechanisms, primarily involving inhibitory lymphocytes and other immune cells that release cytokines suppressing cytotoxic CD8 lymphocytes. miR-762, an exosomal microRNA from mice epithelial cells [25]. This study identified that horizontal transmission of miR-762 from OSCC cells may contribute to immune escape by suppressing CXCR3 expression and activating the CXCR3-AKT signaling pathway in T lymphocytes. Chemokines in the TME play a crucial role in cancer progression by promoting tumorigenesis [26]. However, chemokines also recruit adaptive immune cells, enhancing anti-tumor immunity. For instance, CXCR3, a receptor for chemokines CXCL9, CXCL10, and CXCL11, is more highly expressed in tumor tissues than in normal tissues [27]. In cancer cells, autocrine CXCR3 signaling promotes metastasis and growth through AKT signaling [8], while CXCL9 and CXCL11 can inhibit tumor growth by facilitating immune cell infiltration [28]. Thus, tumors must find new ways to evade the immune system. Horizontal gene transfer (HGT) is vital in TME [29]. MicroRNAs HGT quickly inhibits recipient cells gene expression by being continuously supplied from cancer cells. MicroRNAs are easily neutralized by complementary miRNA sequences [30]. After COVID-19 pandemic, RNA-based medicines offer new therapeutic options for disease [31]. Our discovery of miR-762 in OSCC immune escape via horizontal transmission to immune cells in the TME suggests a new niche for RNA-based cancer therapies.

OSCC is typically an immune-cold tumor, showing lower T and NK cell infiltration [32]. CXCR3 is vital for lymphocyte migration and activation, and miR-762 in the TME may help maintain an immunosuppressive environment. In addition to aiding immune escape during tumor initiation, the immune cell content in TME is crucial for predicting treatment outcomes and patient prognosis [33]. miR-762 distribution in TME prevents immune cell infiltration and worsens responses to conventional cancer treatments. Using anti-sense miR-762 could remove miR-762 from the TME, enhance immune cell infiltration, and boost anti-tumor cytotoxicity, improving responses to chemotherapy and radiotherapy.

In OSCC, CXCR3 enhances lymphatic invasion and cancer growth [34], while CXCR3A promotes cancer stemness and chemoresistance [35], indicating CXCR3 is important for OSCC progression. In late-stage NSCLC, CXCR3 expression is diminished in both CD4 and CD8 T cells [36]. CXCR3 is highly expressed in adaptive immune cells, aiding in chemotaxis and activation. Reduced NK-cell CXCR3 expression hinders immune cell recruitment to tumors [37]. Suppressing CXCR3 can promote immune escape. Interestingly, miR-762-treated OSCC-CM increases CXCR3B expression, which acts as the CXCL4 receptor [38]. CXCL4/CXCR3B has opposite effects to CXCL10 in T cells, reducing proinflammatory IFN-gamma and increasing TH2 cytokines [39]. CXCL4 inhibits activated T-cell proliferation [40] and stimulates Treg growth [41]. CXCR3B expressions may enhance OSCC immune escape and immunosuppressive environments via miR-762 transmission. In summary, a novel OSCC escape mechanism through miR-762 transmission, preventing immune cell infiltration and anti-tumor cytotoxicity, may offer new niches for restoring anti-OSCC immunity.

## 4. Materials and Methods

### 4.1. Chemical Reagents, Kits, Antibodies, and Primers

All reagents, kits, antibodies, and primer sequences are listed in Appendix A. CXCR3 vectors were created by Gateway cloning, as described in our previous study [42]. Lentiviral particles were generated by Taiwan RNAiCore standard protocol [43]. The target cells were infected and antibiotic selection. The miR-762-PM transfection mix was prepared using TransIT-X2 according to the manufacturer’s protocol and applied to Jurkat cells.

### 4.2. Cell Culture Conditions

Human acute T-cell leukemia Jurkat cells., human embryonic kidney 293T, and human oral cancer cells SCC-4, SCC-9, SCC-15, SCC-25, Cal-27, Cal-33, HSC-2, HSC-3, HSC-3-M3, and HSC-4, Ca9-22, OSC-19, OSC-20, and SAS were purchased from the original resource and listed in Appendix A. All cells were cultured in the standard culture condition by following the instruction manual and maintained for 3 months. Additionally, all cells were regularly checked for mycoplasma infection and cell morphology to ensure cell health.

### 4.3. OSCC Conditioned Medium Preparation

Conditioned medium from OSCC was prepared using Cal-27 cells transfected with either miR-762-PM or control microRNA (Genomics, New Taipei City, Taiwan). Briefly, 50 nM of microRNA was transfected into Cal-27 cells using TransIT-X2 reagents (Mirus Bio, Madison, WI, USA). After 16 h of incubation, the transfected cells were washed and replaced with serum-free medium, followed by an additional 24 h incubation. The conditioned medium was then collected, filtered through a 0.22 μm PVDF filter to remove cell debris, and stored at −80 °C until further use. OSCC exosomes were harvested from ultra-centrifuge purification methods [44].

### 4.4. Activation of Jurkat T-Cell

The Jurkat cells were stimulated with 1.25 μg/mL of anti-CD3 (Clone: OKT3; BioLegend, San Diego, CA, USA) monoclonal antibody (mAb) and 1.25 μg/mL of anti-CD28 (Clone: CD28.2; BioLegend) mAb for 72 h. Jurkat-CXCR3 cells were stimulated with the same amount of anti-CD3 and anti-CD28 mAb plus 10 ng/mL of IL-12 (R&D Systems, Inc., Minneapolis, MN, USA) for 72 h.

### 4.5. T-Cell Cytotoxicity Assay Against OSCC Cells

In this study, 2 × 10^4^ Jurkat cells were seeded in the upper-Transwell chamber (Wuxi NEST Biotechnology, Wuxi, China) and co-transfected with or without miR-762 mimics and CXCR3-CDS expression vector for 72 h. For T-cell cytotoxicity, 1 × 10^4^ Cal-27 cells were seeded in the lower well. After 48 h, Cal-27 cells were fixed in 70% ice-cold ethanol for 60min, then stained with a 0.1% crystal violet/20% methanol solution and recorded under a light microscope.

### 4.6. Gene Expression Analysis

Total RNA was isolated by TRIzol reagent (Thermo Fisher Scientific, Waltham, MA, USA). cDNA was synthesized by Roche First Strand cDNA Synthesis Kit (Roche, Basel, Switzerland). All primers were listed in Appendix A. qRT-PCR analysis was used with ChamQ Universal SYBR qPCR Master Mix, (Vazyme Biotech, Nanjing, China), Gunster Biotech MB-Q96-LP qPCR plate (Gunster Biotech, New Taipei City, Taiwan) and QuantStudio 3 qPCR system Thermo Fisher Scientific, Waltham, MA, USA). For miRNA detection, cDNA was prepared by the QIAGEN miScript II RT Kit (Qiagen, Hilden, Germany). Antibodies were listed in Appendix A. Furthermore, 10–30 μg protein lysates were loaded onto % SDS-PAGE and blotted to PVDF membranes. The protein expression was analyzed by the e-BLOT Touch Imager (e-BLOT, Shanghai, China). Supernatants were collected and analyzed by IL-12 ELISA kits (R&D System, Minneapolis, MN, USA).

### 4.7. T-Cell Cytotoxicity and Migration Assays

In this study, 2 × 10^4^ Jurkat cells were seeded in the 0.4-μm upper chamber and co-transfected with miR-762-PM and CXCR3-CDS vector for 72 h. For the T-cell cytotoxicity assay, 1 × 10^4^ Cal-27 cells were seeded in the lower well. For migration assays, Jurkat cells were seeded in the 3 μm upper chamber and incubated with different lower chamber conditions for 24 h. The cytotoxicity of OSCC cells (48 h) or migrated Jurkat cells was fixed, stained, and recorded by microscope (200× magnification, Leica, Wetzlar, Germany). miR-762-PM-transfected Jurkat cells were seeded and incubated for 72 h by CCK-8 assay. Moreover, 10% CCK-8 (dojindo, Kumamoto, Japan) was added and incubated for an extra 3 h, then measured OD450 by Thermos Multiskan reader (Thermo Fisher Scientific, Waltham, MA, USA).

### 4.8. Reporter Assays for Transcription and 3′UTR Reporter Activity

For the NF-kB and CDH1 activity assay, the experiment was performed in Cal-27 cells. For the 3′-UTR reporter assays, the CXCR3 3′-UTR region was cloned by PCR amplification (446 bp after the CXCR3 3′-UTR stop codon) and cloned into the modified CpoI sites of the pCMV-Greenfire-3′-UTR vector (System Biosciences, Palo Alto, CA, USA). The pGF-CXCR3-UTR miR-762 binding site mutation clone was generated by site-directed mutagenesis with the primer listed in Appendix A. The CXCR3 3′UTR-reporter-expressed Cal-27 cells were seeded into 24-well plates and co-transfected with 25 nM miR-762 mimics or NC control. All CXCR3-UTR reporters were packaged into pseudovirus particles and introduced into Cal-27 cells. After 48 h, the 3′-UTR reporter activity was measured by the ONE-Glo™ substrate (Promega, Madison WI, USA) and SpectraMaxID3 reader (Molecular Device, San Jose, CA, USA).

### 4.9. Flow Cytometry Analysis

To determine surface markers, cells were labeled with the surface CD25 and CD69 for 60 min on ice in the dark. The stained cells were analyzed using an Attune NxT Flow Cytometer (Thermo Fisher Scientific, Waltham, MA, USA), and the quantitative analysis was performed using FlowJo v10 (Becton-Dickson, Franklin Lakes, NJ, USA)

### 4.10. Statistical Analysis

All experiments were repeated at least 3 times. Data were presented as mean ± standard deviation (SD). Differences between various treatment groups were assessed using Student’s *t* test. Between-group differences were considered significant at *p* < 0.05. Data analyses were performed using GraphPad Prism version 8 (Dotmatics, Boston, MA, USA). 

## Figures and Tables

**Figure 1 ijms-26-01077-f001:**
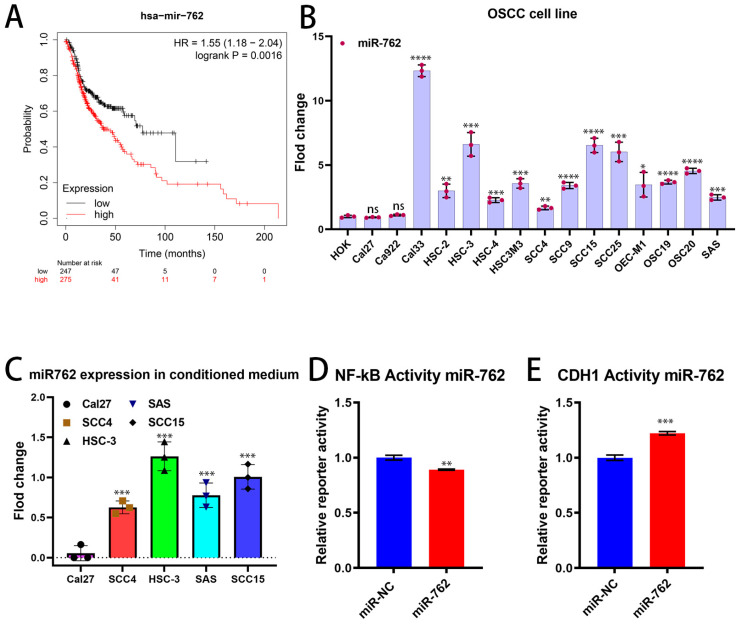
miR-762 is expressed and secreted by OSCC cells. (**A**) Overall survival time was significantly associated with miR-762-low grade compared to miR-762-high grades in 522 OSCC samples from the Kaplan–Meier Plotter (https://kmplot.com/analysis/ accessed on: 2 May 2024). (**B**,**C**) qRT-PCR analysis to validate the expression levels of miR-762 in OSCC cell lines and OSCC cell line-conditioned medium. The U6 small nuclear RNA was used as an internal normalized control. (**D**) NF-kB transactivation assays. (**E**) CDH1 (E-cadherin) promoter activity assay was performed in Cal-27 cells. The empty luciferase vector was used as a background control. Values represent mean ± SD (*n* = 3). Significance levels were determined using Student’s *t*-test (ns: non-significant, * *p* < 0.05, ** *p* < 0.01, *** *p* < 0.001, and **** *p* < 0.0001).

**Figure 2 ijms-26-01077-f002:**
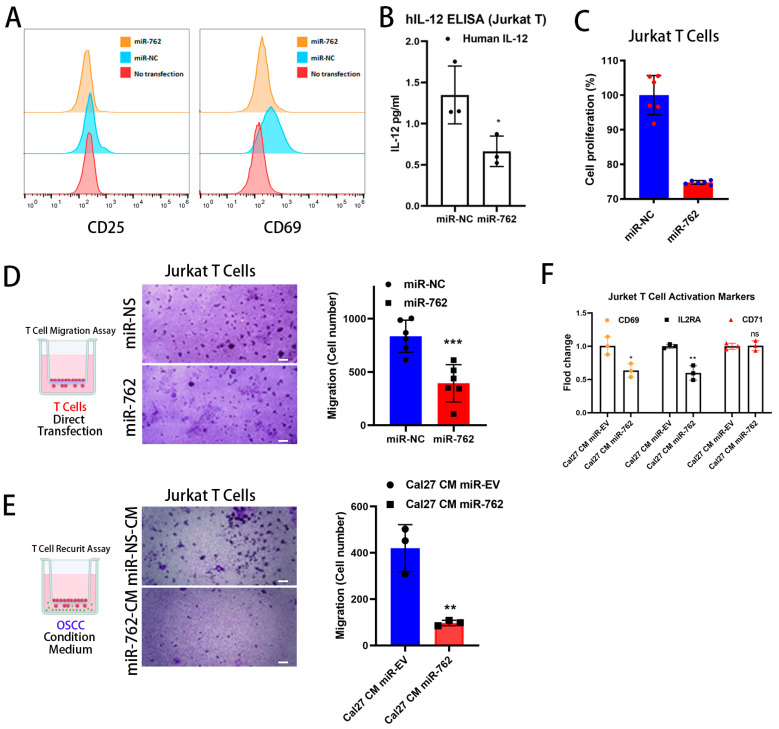
The overexpression of miR-762 inhibited the activation of Jurkat cells. (**A**) Jurkat cells were transfected with miR-762 or control, co-stimulated with anti-CD3/anti-CD28 antibodies for 72 h, and cell activation was detected by flow cytometry. (**B**) ELISA analysis of IL-12 in the medium of Jurkat cells was transfected with miR-762 mimics or N.C. miRNA for 72 h. Data represented means ± SD from three independent experiments (each experiment contains two technical replicates). (**C**) Jurkat cells were transfected with miR-762 or N.C. miRNA for 72 h, and cell proliferation was determined by the CCK assay. * *p* < 0.05 versus the N.C. group. (**D**) Jurkat cells were transfected with miR-762 mimics or non-targeting control miRNA for 72 h before seeding in the Transwell for 24 h. Scale bar indicates 100 µm. (**E**) Jurkat cells were transfected with miR-762 mimics or non-targeting control miRNA (N.C. miRNA) for 72 h, and the expression levels of IL-2RA, CD69, and CD71 were detected by qRT–PCR. GAPDH was used as an internal control. Scale bar indicates 100 µm. (**F**) Representative images and quantified bar charts of Transwell migration assays of Jurkat cells treated with a conditioned medium from Cal-27 cells overexpressing miR-762 (miR-762-CM) or control vector-transfected Cal-27 cells (miR-NS-CM) for 24 h. Values represent mean ± SD (*n* = 3), with N = 2 replicates. Significance levels were determined using Student’s *t*-test (ns: non-significant, * *p* < 0.05, ** *p* < 0.01, and *** *p* < 0.001).

**Figure 3 ijms-26-01077-f003:**
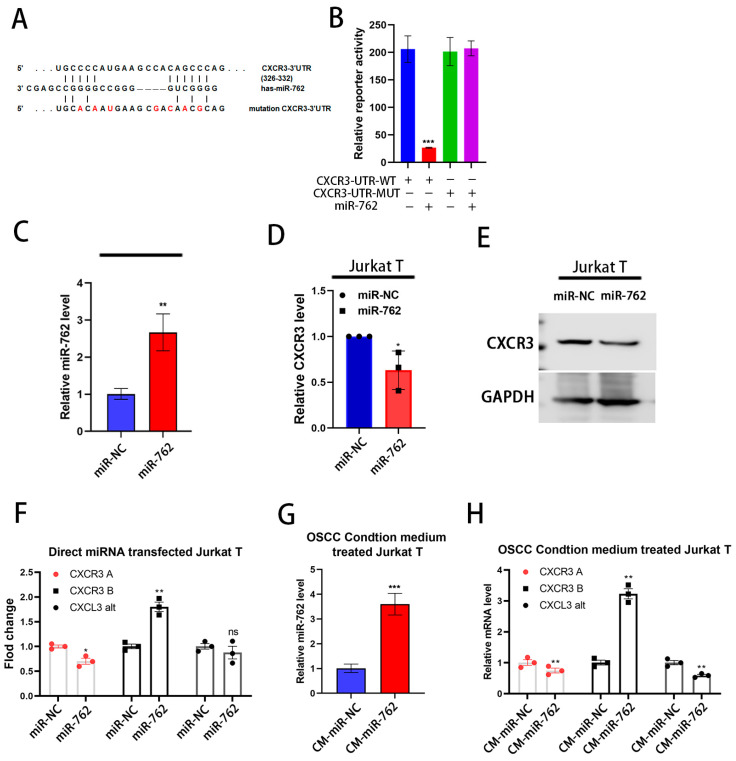
CXCR3 is a direct target of miR-762. (**A**) Schematic representation of the putative miR-762 binding sequence in the 3′-UTR of CXCR3 with the wild-type form (CXCR3 3′-UTR WT) and the mutation CXCR3 3′UTR-reporter clone. (**B**) Luciferase activities were measured to assess the effect of miR-762 mimics (PM, 50 nM) on constructs containing the wild-type or the mutation CXCR3 3′-UTR in Cal-27 cells. The relative luciferase activity of each sample was measured at 48 h after transfection and normalized to Renilla luciferase activity. (**C**,**D**) qRT–PCR analysis showing the expression levels of miR-762 (**C**) and CXCR3 (**D**). (**E**) Western blot analysis of CXCR3 after transfection of miR-762 mimics with 50 nM for 72 h in Jurkat cells. GAPDH was used as an internal control. (**F**) the CXCR3 variant analysis from CXCR3 A, CXCR3 alt, and CXCR3 B in Jurkat T cell lines after transfection with 50 nM of miR-762 mimics (PM). (**G**,**H**) The miR-762 expression (**G**) and CXCR3 variant analysis (**H**) from Jurkat T cells treated with conditioned medium from Cal-27 cells overexpressing miR-762 (CM-miR-762) or control vector-transfected Cal-27 cells (CM-miR-NC) for 72 h. GAPDH was used as an internal control. Significance levels were determined using Student’s *t*-test (ns: non-significant, * *p* < 0.05, ** *p* < 0.01, and *** *p* < 0.001).

**Figure 4 ijms-26-01077-f004:**
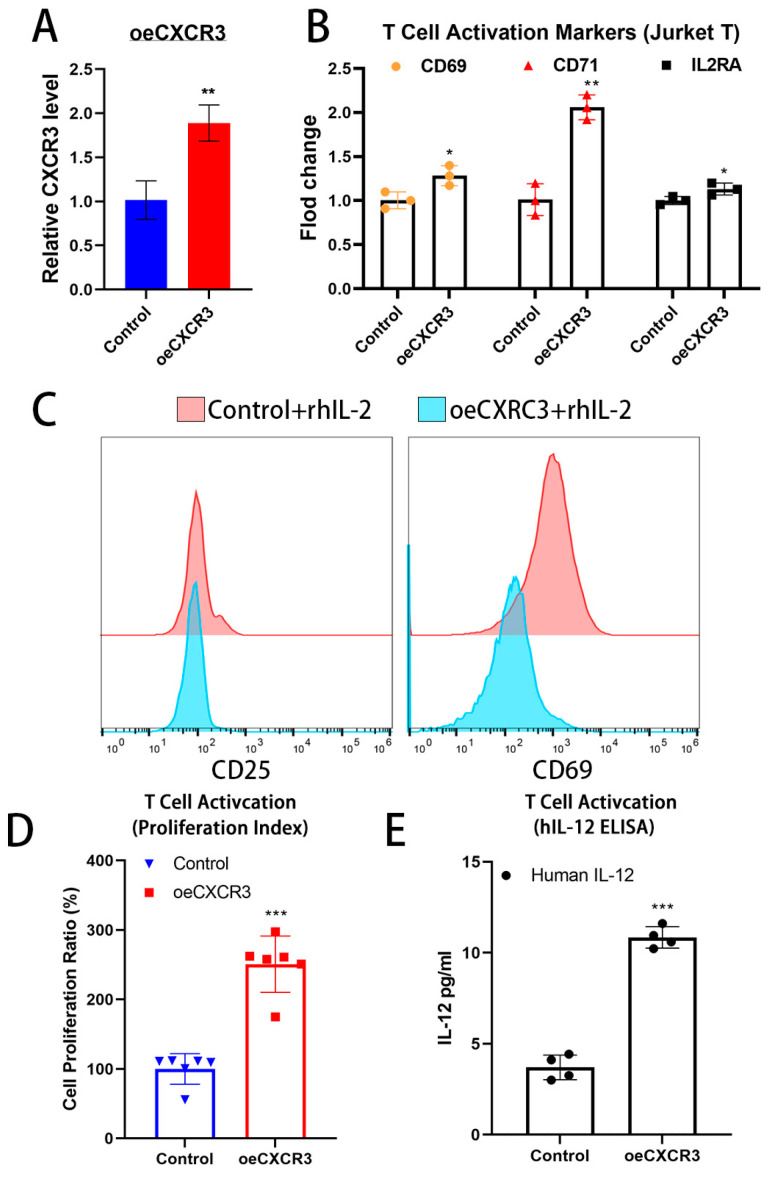
CXCR3 induced the Jurkat T-cell activation and proliferation. (**A**) qRT-PCR analysis was conducted to measure the expression levels of CXCR3. (**B**) qRT-PCR analysis of T-cell activation genes after stably expressing CXCR3. GAPDH was utilized as an internal control. (**C**) CXCR3-expressed Jurkat cells were then co-stimulated with IL-2 (10 ng/mL) for 72 h, and cell activation was assessed by flow cytometry. (**D**) Cell proliferation ability of CXCR3-expressed Jurkat cells. Cell proliferation index was measured using the CCK-8 assay for 72 h induction. (**E**) ELISA analysis of IL-12 in the CXCR3-expressed Jurkat condition medium. Data represent means ± SD from three independent experiments, with each experiment containing two technical replicates. Significance levels were determined using Student’s *t*-test (* *p* < 0.05, ** *p* < 0.01, and *** *p* < 0.001).

**Figure 5 ijms-26-01077-f005:**
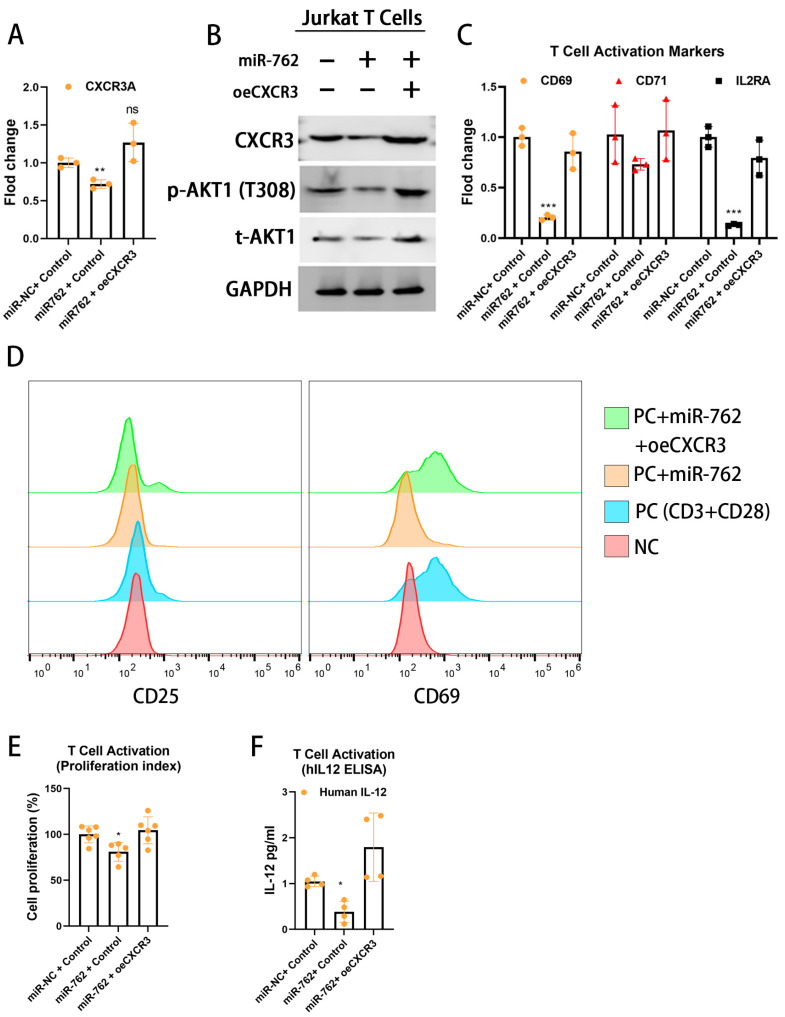
Horizontal miR-762 inhibits cell activation by targeting CXCR3. Jurkat cells were co-transfected with miR-762 mimics and CXCR3 CDS for 72 h. (**A**) CXCR3 mRNA level. (**B**) CXCR3 protein level and downstream phosphor-Akt activity. Numerical values for protein band intensities are depicted below the gels. The values were quantitated by densitometry and normalized to GAPDH. (**C**) qRT-PCR analysis of T-cell activation genes. GAPDH was used as an internal control. (**D**) Flow cytometry analysis of surface CD69 and CD25 expression on miR-762 and CXCR3-expressed Jurkat cells was followed by co-stimulation with anti-CD3/anti-CD28 antibodies for 72 h. (**E**) Jurkat T-cell proliferation ability (**F**). IL12-ELISA analysis in the Jurkat cells medium with miR-762 mimics and CXCR3-CDS expression for 72 h. Significance levels were determined using Student’s *t*-test (ns: non-significant, * *p* < 0.05, ** *p* < 0.01, and *** *p* < 0.001).

**Figure 6 ijms-26-01077-f006:**
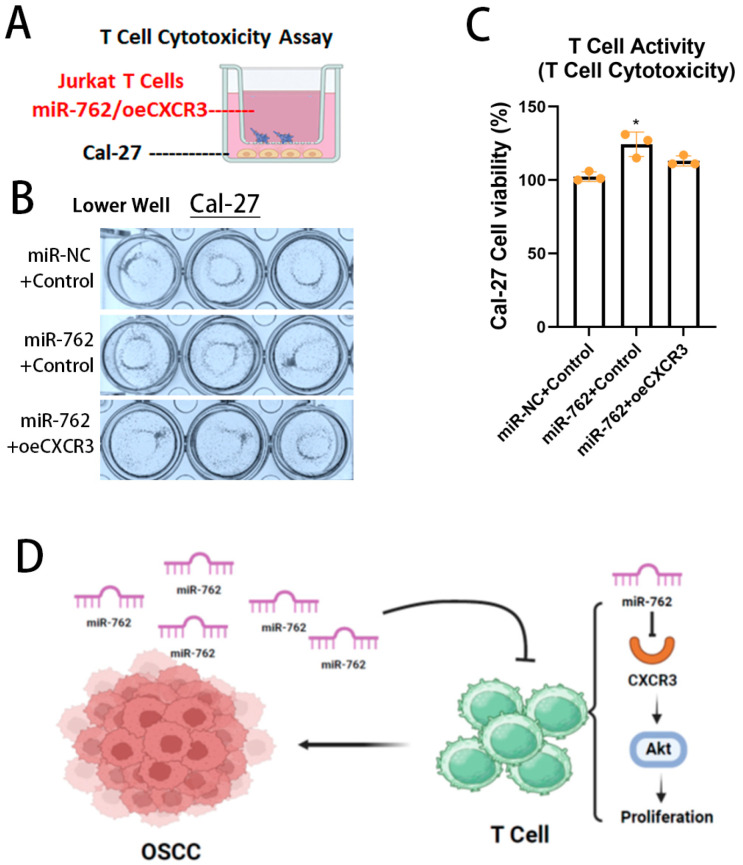
miR-762 attenuates Jurkat T-cell antitumor cytotoxicity. (**A**) Schematic diagram of Jurkat T-cell-induced OSCC cytotoxicity assay. (**B**,**C**) Representative images (**B**) and quantified bar charts of CAL-27 cell viability (**C**) of T-cell antitumor cytotoxicity with miR-762 and CXCR3-CDS expression. Values represent mean ± SD (*n* = 3). (**D**) OSCC cells inhibit T-cell activation through inhibition of CXCR3 by horizontal miR-762 transmission. Significance levels were determined using Student’s *t*-test (* *p* < 0.05).

## Data Availability

The original contributions presented in this study are included within this article/Appendix A. Further inquiries can be directed to the corresponding author.

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
