# Peer review of "Oral Cancer-Derived miR-762 Suppresses T-Cell Infiltration and Activation by Horizontal Inhibition of CXCR3 Expression"

_ijms, 2025, doi:10.3390/ijms26031077_

Round 1
Reviewer 1 Report
Comments and Suggestions for Authors
The manuscript by Peng et al. explores a novel miR-762/CXCR3 axis that modulates the immunosuppressive microenvironment in OSCC and could serve as a potential RNA-targeted therapeutic strategy to enhance the anti-tumor immune response in the treatment of OSCC.
I have a few suggestions for polishing the commentary.
Comments
- The authors should include statistics in Fig. 1B and C.
- The authors should further dissect how miR-762 expression suppresses CXCR3 mRNA and protein expression. Does it impact the stability and/or transcription of the mRNA, or the stability and/or translation of the protein?
- Did the authors check the expression of Wnt target genes in the presence and absence of miR-762?
- Did the authors find this unique signature miR-762 as an exosomal vesicle derivative in OSCC? Did the authors identify any other signature miRNAs?
- Is the current model restricted to only cell lines? Did the authors test their hypothesis in mouse models?
Author Response
-
Comments
- The authors should include statistics in Fig. 1B and C.
Ans: Thank you for the reviewer’s kind comment. We have added the statistics in Fig. 1B and C. We also correct the Figure 1 legend.
- The authors should further dissect how miR-762 expression suppresses CXCR3 mRNA and protein expression. Does it impact the stability and/or transcription of the mRNA, or the stability and/or translation of the protein?
Ans: We thank the reviewer for their valuable comment. Based on the results presented in Figure 3C, we observed that miR-762 directly suppresses mRNA expression levels. In mammalian cells, microRNAs typically regulate gene expression through two mechanisms: causing mRNA instability and inhibiting protein translation. However, not all microRNAs exhibit mRNA suppression. From our observations, the miR-762-induced suppression of CXCR3 suggests that miR-762 affects both mRNA stability and protein expression of CXCR3.
To further clarify the regulatory mechanism of miR-762 on CXCR3, we have added the following sentence on lines 167 to 170 page 5:
"Additionally, ectopic miR-762 expression directly suppressed CXCR3 mRNA and protein expression in Jurkat cells (Figure 3C to E). These results indicate that miR-762 regulates CXCR3 through both reducing mRNA stability and inhibiting protein translation."
- Did the authors check the expression of Wnt target genes in the presence and absence of miR-762?
Ans: Thank you for the reviewer’s critical comments. We have not yet examined the expression of WNT target genes upon miR-762 expression in OSCC. While WNT signaling is known to play a significant role in oral cancer tumorigenesis and progression, as highlighted in our previous study and other references, this manuscript focuses on the immune suppression mechanism of oral cancer, particularly through the prevention of T cell recruitment.
The original study hypothesized that miR-762 serves as an oncogenic microRNA in OSCC progression, similar to its role in CRC, by enhancing WNT pathway activation. In our preliminary analysis, we found that miR-762 indeed promotes the in vitro growth of Cal-27 cells. However, when tested in vivo using Nod-SCID mice, miR-762 was observed to suppress Cal-27 tumor growth. Furthermore, miR-762 has been predicted to target genes such as JAG1, NOTCH1-3, SHH, PTCH1, and FZD7, which are associated with key pathways. Our results confirm that miR-762 suppresses the expression of these genes and inhibits downstream pathway activation, which may also influence immune cell activation.
Taken together, these findings suggest that miR-762 may play a critical role in modulating the tumor microenvironment. This provides additional insights into its potential horizontal role in mediating interactions between OSCC and immune cells, warranting further investigation.

Since these results are solely derived from OSCC cells and represent a promising direction for future studies, we acknowledge that including them in the manuscript may confuse the rationale of the original model. Therefore, we have chosen to present these findings exclusively in the rebuttal letter.
Thank you again for your valuable and insightful comments, which have greatly contributed to refining our study and guiding future research directions.
- Did the authors find this unique signature miR-762 as an exosomal vesicle derivative in OSCC? Did the authors identify any other signature miRNAs?
Ans: Thank you for the critical comment. Indeed, we identified miR-762 as an exosomal microRNA derived from OSCC. To confirm this, we harvested the supernatant and purified exosomes using ultracentrifugation methods, demonstrating that miR-762 can be detected in OSCC-derived exosomes.
We have included these findings in the supplementary figure and added a corresponding description in the Results section. Please refer to lines 103–104 on page 3:
"miR-762 serves as an exosomal microRNA from OSCC cells (Supplementary Figure 1)."
- Is the current model restricted to only cell lines? Did the authors test their hypothesis in mouse models?
Ans: Thank you for the reviewer’s critical comments. Regarding the role of miR-762 in preventing T cell recruitment in oral cancer, our current study demonstrates this model at the cellular level. However, due to the lack of an immune-competent OSCC mouse model, we are unable to directly validate the interaction between cancer cells and immune cells in vivo. We sincerely thank the reviewer for their insightful and critical comments, which have greatly helped us improve the clarity and focus of our study. We plan to validate this model in the future using an immune-competent mouse model, such as the C57B6/MTCQ1 model. While this is an important direction for our research, we acknowledge that we are unable to provide conclusive in vivo data at this time. We deeply appreciate the reviewer’s understanding of this limitation and their constructive feedback, which will guide us in our future efforts.

Reviewer 2 Report
Comments and Suggestions for Authors
In this manuscript, the authors revealed that OSCC-derived miR-762 suppressed T cell activation. Mechanistically, transmitted miR-762 targeted CXCR3 and repressed the downstream pathway related to proliferation in T cell.
The manuscript is interesting, but there are some concerns that the authors need to clarify. Listed below are my specific comments.
1. Is miR-762 an EV-contained microRNA or a cell-free microRNA? The authors should discuss this matter in the Discussion section.
2. Did the authors measure the proliferation change of Cal27 cells after miR-762 transfection? Which genes are targeted by miR-762 in OSCC cells?
3. Fig.1D, E
Which cells did the authors use in the luciferase assay (NF-kB and CDH1)? They also should explain this luciferase assay procedure in the Materials and Methods section.
4. Fig.3A
Which prediction tool did the authors use to show the possible interaction between CXCR3 and miR-762?
5. Fig.3B and Lines 155-156
It is mandatory to use the mutated reporter at the putative binding site to demonstrate the direct interaction between miRNA and its target.
6. Fig.3F, H
Why was CXCR3B increased? The authors should explain the possible mechanisms.
7. Line 324
The concentration (%) of SDS-PAGE was missing.
8. The authors should explain how they prepared the conditioned medium in the Materials and Methods.
9. Abbreviation should be defined at first mention (e.g., Line 34 microRNA, Line 76 EVs, Line 120 miR-762-PM, Line 180 CXCR3 alt, etc.).
Author Response
In this manuscript, the authors revealed that OSCC-derived miR-762 suppressed T cell activation. Mechanistically, transmitted miR-762 targeted CXCR3 and repressed the downstream pathway related to proliferation in T cell.
The manuscript is interesting, but there are some concerns that the authors need to clarify. Listed below are my specific comments.
- Is miR-762 an EV-contained microRNA or a cell-free microRNA? The authors should discuss this matter in the Discussion section.
Thank you for the critical comment. Indeed, we identified miR-762 as an exosomal microRNA derived from OSCC. To confirm this, we harvested the supernatant and purified exosomes using ultracentrifugation methods, demonstrating that miR-762 can be detected in OSCC-derived exosomes.
We have included these findings in the supplementary figure and added a corresponding description in the Results and discussion section. Please refer to lines 103–104 on page 3:
"miR-762 serves as an exosomal microRNA from OSCC cells (Supplementary Figure 1)."
And lines 288 on page 10:
miR-762, an exosomal microRNA from mice epithelial cells

- Did the authors measure the proliferation change of Cal27 cells after miR-762 transfection? Which genes are targeted by miR-762 in OSCC cells?
Ans: Thank you for the reviewer’s critical comments and thoughtful questions.
- Proliferation Change of Cal-27 Cells after miR-762 Transfection:
In our preliminary analysis, we observed that miR-762 promotes the in vitro proliferation of Cal-27 cells following transfection. This finding aligns with the hypothesis that miR-762 may serve as an oncogenic microRNA in OSCC progression. However, interestingly, when tested in vivo using Nod-SCID mice, miR-762 was observed to suppress Cal-27 tumor growth. This discrepancy between in vitro and in vivo results highlights the complexity of miR-762’s role and suggests that its effects may depend on the tumor microenvironment or interactions with immune components. - Genes Targeted by miR-762 in OSCC Cells:
Bioinformatic analyses and experimental validation indicate that miR-762 targets several key genes, including JAG1, NOTCH1-3, SHH, PTCH1, and FZD7. These genes are involved in critical signaling pathways such as the WNT, NOTCH, and Hedgehog pathways, which are well-documented to influence tumorigenesis and immune modulation. Our findings confirm that miR-762 suppresses the expression of these target genes, thereby inhibiting downstream pathway activation. This suppression could also play a role in modulating immune cell activation within the tumor microenvironment.

While these observations are valuable, this manuscript primarily focuses on the immune suppression mechanism of oral cancer, specifically the role of miR-762 in preventing T cell recruitment. Although WNT signaling and other pathways are significant in oral cancer progression, examining their involvement in miR-762-mediated effects lies beyond the scope of this study.
Since these results are derived from OSCC cells and represent a promising area for future research, we have chosen to include them in the rebuttal letter rather than the main manuscript to avoid confusing the rationale of the original model. Future studies will focus on comprehensively characterizing the role of miR-762 in tumorigenesis and its potential interactions with immune cells within the tumor microenvironment.
Thank you again for your valuable and insightful comments, which have greatly contributed to refining our study and guiding future research directions.
Which cells did the authors use in the luciferase assay (NF-kB and CDH1)? They also should explain this luciferase assay procedure in the Materials and Methods section.
Ans: We thank the reviewer for their comment. We used Cal-27 cells for the NF-kB and CDH1 luciferase activity assays. This information has been added to the Materials and Methods section for clarity.
Please refer to line 381 on page 12 or see the description below:
"For the NF-kB and CDH1 activity assays, the experiment was performed in Cal-27 cells."
Which prediction tool did the authors use to show the possible interaction between CXCR3 and miR-762?
Ans: We thank the reviewer for their comment. We used TargetScan to predict the binding interaction between miR-762 and CXCR3. This information has been added to the Results section for clarity.
Please refer to lines 163–164 on page 5 or see the description below:
"TargetScan analysis predicted a direct binding interaction between miR-762 and CXCR3."

It is mandatory to use the mutated reporter at the putative binding site to demonstrate the direct interaction between miRNA and its target.
Ans: We thank the reviewer for their insightful comment. To demonstrate the direct interaction between miR-762 and its target, we generated a mutated pGF-CXCR3-UTR miR-762 binding site clone using site-directed mutagenesis. All CXCR3-UTR reporters were packaged into pseudovirus particles and introduced into Cal-27 cells. The binding pattern of the mutated CXCR3-UTR with miR-762 was predicted using mFold double strain RNA binding prediction.
The updated results have been included in Figure 3B for clarity, along with a description in the Results section and modify in material and method section. Please refer to lines 164–167 or see the excerpt below: (Results)
"We performed a CXCR3-3’-UTR reporter assay with or without miR-762 binding site mutation. The CXCR3-3’-UTR was significantly suppressed by miR-762 mimic (PM) transfection but was abolished by the miR-762 binding site mutation (Figure 3B)."

And also refer to lines 384–390 or see the excerpt below: (Methods)
The pGF-CXCR3-UTR miR-762 binding site mutation clone was generated by site-direct mutagenesis with primer listed in supplementary table S1. The CXCR3 3’UTR-reporter expressed Cal-27 cells were seeded into 24-well plates and co-transfected with 25 nM miR-762 mimics or NC control. All CXCR3-UTR reporters were packaged into pseudo-virus particles and introduced into Cal-27 cells. After 48 hours, the 3’-UTR reporter activity was measured by ONE-Glo™ substrate and SpectraMaxID3 reader.
Why was CXCR3B increased? The authors should explain the possible mechanisms.
Ans:
We thank the reviewer for raising this important question. CXCR3A and CXCR3B are regulated by alternative splicing, which can lead to differential expression and function of these isoforms under specific conditions. Unlike CXCR3A, CXCR3B is regulated by unique transcription factors, NRF2 and BACH1, which play a critical role in modulating its expression and downstream effects.
As stated in lines 177–185 on page 5:
" Both direct transfection of miR-762 and OSCC-derived miR-762 conditioned medium were found to stimulate the expression of the CXCR3B variant, which is generated through alternative splicing (Figure 3F and H). Unlike CXCR3B, which exhibits tumor-suppressive functions, CXCR3A promotes tumorigenesis by enhancing cell proliferation and migration. The regulation of CXCR3B involves unique transcription factors, NRF2 and BACH1 , which modulate its expression and downstream effects in response to miR-762. This suggests that miR-762 may influence the alternative splicing or transcriptional regulation of CXCR3 isoforms, thereby shaping their distinct roles in tumor progression and immune modulation. "
These findings suggest that miR-762 may indirectly influence CXCR3B expression through its interaction with NRF2 and BACH1, resulting in the observed increase.
- Line 324
The concentration (%) of SDS-PAGE was missing.
Ans: Thank you for your kind instructions. The missing 10% has been added to the Materials and Methods section for completeness and clarity.
- The authors should explain how they prepared the conditioned medium in the Materials and Methods.
Ans: Thank you for your kind instructions and we have add conditioned medium prepared methods in the material and method section. Please referred to lines 341 to 348, page13 or see in the below:
OSCC-conditioned medium preparation
Conditioned medium from OSCC was prepared using Cal-27 cells transfected with either miR-762-PM or control microRNA. Briefly, 50 nM of microRNA was transfected into Cal-27 cells using TransIT-X2 reagents. After 16 hours of incubation, the transfected cells were washed and replaced with serum-free medium, followed by an additional 24 hours of incubation. The conditioned medium was then collected, filtered through a 0.22 μm PVDF filter to remove cell debris, and stored at -80°C until further use.
- Abbreviation should be defined at first mention (e.g., Line 34 microRNA, Line 76 EVs, Line 120 miR-762-PM, Line 180 CXCR3 alt, etc.).
Ans: Thank you for your instructions. All abbreviations, including those mentioned (e.g., microRNA, EVs, miR-762-PM, CXCR3 alt), have been defined at their first mention in the manuscript for clarity and consistency.

Round 2
Reviewer 1 Report
Comments and Suggestions for Authors
The manuscript by Peng et al. explores a novel miR-762/CXCR3 axis that modulates the immunosuppressive microenvironment in OSCC and could serve as a potential RNA-targeted therapeutic strategy to enhance the anti-tumor immune response in the treatment of OSCC.
The authors have addressed all the previous comments. Thus, the manuscript can be accepted in its present form.
Reviewer 2 Report
Comments and Suggestions for Authors
I am satisfied with the revisions that have been made by the authors. I recommend this manuscript for publication.